# High Linearity Synaptic Devices Using Ar Plasma Treatment on HfO_2_ Thin Film with Non-Identical Pulse Waveforms

**DOI:** 10.3390/nano12183252

**Published:** 2022-09-19

**Authors:** Ke-Jing Lee, Yu-Chuan Weng, Li-Wen Wang, Hsin-Ni Lin, Parthasarathi Pal, Sheng-Yuan Chu, Darsen Lu, Yeong-Her Wang

**Affiliations:** 1Program on Semiconductor Process Technology, Academy of Innovative Semiconductor and Sustainable Manufacturing, National Cheng-Kung University, Tainan 701, Taiwan; 2Department of Electrical Engineering, Institute of Microelectronics, National Cheng-Kung University, Tainan 701, Taiwan; 3Department of Physics, National Sun Yat-sen University, Kaohsiung 804, Taiwan

**Keywords:** electrical synapse, synaptic plasticity, argon plasma treatment, non-identical pulse waveforms

## Abstract

We enhanced the device uniformity for reliable memory performances by increasing the device surface roughness by exposing the HfO_2_ thin film surface to argon (Ar) plasma. The results showed significant improvements in electrical and synaptic properties, including memory window, linearity, pattern recognition accuracy, and synaptic weight modulations. Furthermore, we proposed a non-identical pulse waveform for further improvement in linearity accuracy. From the simulation results, the Ar plasma processing device using the designed waveform as the input signals significantly improved the off-chip training and inference accuracy, achieving 96.3% training accuracy and 97.1% inference accuracy in only 10 training cycles.

## 1. Introduction

Big data computing architecture has become more and more popular with the increasing demand of Internet of Things (IoT) [1]. The limited throughput between CPU and memory inhibits the efficiency of big data processing and pushed towards the von Neumann bottleneck [2]. The concept of the artificial neural network (ANN) was developed to achieve efficient computing by mimicking the human brain and neurons [3]. The human brain is a powerful information processing system composed of approximately 10^11^ neurons and 10^15^ synapses, which form a massive 3D network that can process and transfer information, consuming low power [4]. The hardware implemented neural systems are one of the promising alternatives for highly efficient learning and pattern recognition ability of massive information with an equivalent approach consuming low power [5]. Recently, nonvolatile-memory (NVM)-based neuromorphic computing has become very popular due to its ability to follow the synaptic plasticity of the biological neurons, which are capable of revolutionizing the computer architecture [6,7]. The definition of synaptic plasticity can be divided into long-term depression (LTD) and long-term potentiation (LTP) [8]. Among the various electrical synapses, resistive random-access memory (RRAM) is one of the potential candidates that can mimic the synaptic behavior by applying electrical pulses. The basic RRAM is composed of two layers of metal electrodes and a transition metal oxide (TMO) as the middle layer. The main principle of operation is that the resistance of TMO varies with the applied bias voltage, and the internal resistance value is used to distinguish the signal. When the electric field exceeds the threshold value, the dielectric layer collapses and changes from high resistance to low resistance, a process called Forming. After Forming, a change in resistance occurs, and the process from high resistance to low resistance is called SET, and conversely, the process from low resistance to high resistance is called RESET. In addition to ease of operation, Resistive Random-Access Memory (RRAM) has numerous desirable features, such as an excellent scaling potential, simple structure, low power consumption, fast switching speed, excellent reliability, and compatibility with the CMOS fabrication process [9,10].

For artificial synaptic applications, high linear conductance change is the most important factor designed to achieve efficient neuromorphic computing [11,12]. Several non-ideal factors of RRAM devices have been reported, including nonlinear weight updating and device variation, which are common in synaptic devices, owing to the intrinsic properties of these devices [13]. For efficient neuromorphic computation, linear conductance change is one of the essential requirements to achieve analog synapse devices. In this work, we aimed to improve the linear potentiation and depression characteristics. To achieve these, we proposed two solutions. First, we applied Ar plasma treatment on the surface of the HfO_2_ thin film to increase the surface roughness, which could improve the uniformity of the conducting filament. Second, we improved the linear synaptic characteristics by developing a group of non-identical pulse waveforms.

## 2. Materials and Methods

The schematic configuration of the Al/Ti/HfO_2_/Pt/Ti/SiO_2_/Si structures are shown in Figure 1a. The device was fabricated with a crossbar structure on the Si/SiO_2_ substrate. First, 10 nm/120 nm-thick Ti/Pt (Ti was used as an adhesion layer) were deposited through RF sputtering at room temperature and patterned by UV photolithography flowed by the lift-off process to define the bottom electrode (BE). A 100 nm-thick SiO_2_ isolation layer was deposited by electron beam evaporation (E-beam). A 5 × 5 µm^2^ device area was defined by via the hole region using UV–photolithography and was then etched with a reactive ion etching system (RIE) using CF_4_ to open the device area. Hereafter, a 10 nm thick HfO_x_ resistive switching layer was deposited by an atomic layer deposition (ALD) system at 250 °C with Tetrakis ethylmethylamido hafnium (TEMAHf) and an H_2_O precursor. Argon (Ar) plasma was applied on the surface of HfO_x_ thin film in an RIE chamber with 50 W power at room temperature. Then, the top electrode (TE) region was patterned by the lithography process and 40 nm/25 nm-thick Ti/Al (Al was served as capping layer) were deposited through RF sputtering at room temperature, respectively, followed by lift-off process where the TE were obtained. Finally, BE pad was defined by photo lithography and etched by RIE using CF_4_. The TEM image clearly indicates the thickness of each layer in Figure 1b.

## 3. Results

To verify switching performances, the HfO_2_ thin film surface roughness was investigated. Figure 2a shows the atomic force microscopy (AFM) topographic images of the pristine device and Figure 2b shows the plasma-treated device. The root mean square (RMS) value of the plasma-treated HfO_2_ thin films increased to 0.704 nm compared with that of the pristine HfO_2_ film of 0.547 nm. The higher surface roughness led to an increase in the effective contact area of the Ti/HfO_2_ interface, which would lead to the formation of thicker sub-TiO_x_ interfacial layers [14,15]. The formation of a TiO_x_ interfacial layer in the HfO_x_/Ti interface of the pristine and plasma-treated device was further confirmed by the Ti 2p spectra of the X-ray photoelectron spectroscopy (XPS), as seen in Figure 2c,d. The XPS peak at 454.2 eV confirmed the Ti^0^ valency of the Ti metallic state and the peak around 455.4 eV defined the Ti^2+^ valence state to form the TiO and titanium suboxides, while 459.3 eV corresponded to the formation of TiO_2_ bonding with the Ti^4+^ valence state at the switching layer and Ti buffer layer interface [16,17]. It is clearly shown that the intensity significantly increased after plasma treatment due to the formation of the thick TiO_2_ layer. With the increase in sub-TiO_x_ thickness, the oxygen exchange at the interface will be more beneficial, which will result in a higher oxygen vacancy concentration at the interface, and a lower VF with more uniform confinement of conduction filaments (CFs). In order to investigate the effects of Ar plasma on the devices, we first conducted electrical performance measurements and compared the results of the pristine device with the plasma-treated device. The current–voltage (I–V) curves were characterized by an Agilent B1500A parameter analyzer. The BE was grounded, and bias was applied on the TE. Figure 3a,b illustrates the forming curve of the pristine and plasma-treated devices, respectively. It can be observed that the forming voltage (V_F_) of the plasma-treated device improved to 2 V, which is much lower than the pristine device (V_F,pristine_ = 6.05 V). Moreover, the SET voltage (V_SET_) value of the plasma-treated devices (V_SET,plasma_ = 1.05 V) also improved in Figure 3c compared with the pristine device (V_SET,pristine_ = 1.2 V), as shown in Figure 3d. For a further analysis regarding the Ar plasma effect on the memory properties of the devices, we performed endurance and retention reliability tests. Both the DC endurance and retention tests were performed at room temperature at a read voltage (V_read_) of 0.1 V in both the high-resistance state (HRS) and low-resistance state (LRS). In the DC endurance test in Figure 4a, the ON/OFF ratios of both devices were maintained well after 400 DC duty cycles. In addition, both devices showed a retention time longer than 10^4^ s, and without a significant ON/OFF ratio degradation, as shown in Figure 4b.

However, both reliability results indicated enhanced ON/OFF ratio in the plasma-treated device compared with the pristine device. Moreover, we also measured the pulse endurance switching operation (P/E endurance) using a pulse width (PW) of 1 µs. Figure 5 shows the P/E endurance results of the pristine and plasma-treated devices, respectively, with the temperature at a read voltage (V_read_) of 0.1 V in both HRS and LRS during the pulse endurance test. Both devices were able to maintain the ON/OFF ratio at 10^2^. However, the plasma-treated device could successfully maintain the ON/OFF ratio after 10^6^ pulse duty cycles compared with the 10^4^ cycles in the pristine device.

The conduction mechanism was investigated for the fabricated cross-point resistive device. The higher slope value in the ln (*I*)–ln (*V*) curve (not shown) expected the possible Ohmic, or space charge limited current conduction (SCLC), in the HRS. The Schottky emission was validated in the HRS from the linear correlation between the ln (*I*/*T*^2^)–*V*^1/2^ curve in the low field region (<0.7 V), as shown in Figure 5a. The Schottky barrier height (Φ*_SB_*) and the dynamic dielectric constant (ε*_d_*) were determined from the fitted curve using the following equations [18],
(1)$${\;\Phi }_{SB}=\frac{k_{B}T}{q}\left[\ln\left(m^{*}A^{*}\right)-intercept\right]$$
(2)$${\;\varepsilon }_{d}=\frac{q^{3}}{\left[4\pi \epsilon_{0}{\left(slope\right)}^{2}{\left(k_{B}T\right)}^{2}\right]}$$
where *q* is the charge of the electron (1.6 × 10^−19^ C), *T* is the absolute temperature (300 K), *k_B_* is the Boltzmann constant (1.38 × 10^−23^ J/K), ε_0_ is the free space permittivity (8.854 × 10^−14^ F/cm), and *A** is the Richardson–Dushman constant (120 A·cm^–2^ K^2^). The effective mass of the electron (*m**) in HfO_2_ was considered to be 0.15 *m*_0_ [19]. The Schottky barrier height was calculated to be 0.86 eV at the HfO_x_/Pt interface using Equation (1), not including the Fermi-level modulation due to the image charge effect in Figure 6a. Some inconsistency may arise between the theoretically and experimentally calibrated Φ*_SB_*, as described in previous reports [20,21]. In addition, the ε*_d_* of the HfO_x_ layer were extracted to 3.53 using Equation (2). This value was verified by obtaining the optical refractive index (*n*), whose value was 1.87 at 530 nm using the relationship *n*^2^ = ε*_d_* of the ALD deposited HfO_x_ dielectric film near to the prior reported value [22]. The temperature-dependent calibrations ranging from 300 K to 330 K were performed to justify the conduction mechanism. The slope of ln (*I*/*T*^2^)–1/*k_B_T* was linearly dependent so as to justify the Schottky thermionic emission in HRS, as shown in Figure 6b. The effective barrier height (Φ*_eff_*) was calculated in different applied voltages (0.1–0.5 V) using the equation [18]
(3)$$J=A^{*}T^{2}\;exp\frac{-q\left(\Phi_{B}-\sqrt{\frac{qE}{4\pi \varepsilon_{0}\varepsilon_{r}}}\right)}{k_{B}T}$$

Φ*_eff_* was extracted to 0.82 eV, which is close enough to justify the fitting of the conduction mechanism, as shown in Figure 6c. The F-N tunneling conduction was justified in the HRS currents regions by plotting ln (*J*/*E*^2^) vs. 1/*E*, where *E* is the electric field, as shown in the Figure 6d. The tunneling barrier heights (Φ*_b_*) were estimated from the ln (*J*/*E*^2^) vs. 1/*E* plots using the following F-N tunneling equation [18],
(4)$${\;\Phi }_{b}={\left(-S\right)}^{\frac{2}{3}}{\left[\frac{\frac{3}{4}{\left(q\hbar \right)}^{2}}{2m_{ox}}\right]}^{\frac{2}{3}}\;$$
where *S* is the slope, *m_ox_* is the tunneling effective mass of electrons (or holes) in the HfO_x_ layer, and *ħ* is the reduced Planck’s constant. The linear fitting curve validates the F-N tunneling conduction above the critical electric field (*E_C_*) of ≥3 MV/cm [23]. Considering *m_ox_* = 0.15*m*_0_ [19], the value of Φ*_b_* was calculated to 0.76 eV. Therefore, this is indicates that the HRS current at a higher bias of ≥0.9V is dominated by F-N tunneling.

Next, a non-identical pulse waveform was designed to achieve a better linear conductance change, which could emulate the biological synaptic plasticity of long-term depression (LTD) and long-term potentiation (LTP). Several studies have shown that the biological synaptic plasticity of LTD and LTP can be mimicked as a gradual conductance change response to the programming pulses. The identical pulse waveform is commonly proposed to evaluate the synaptic plasticity of the RRAM synapses [24,25]. However, the conductance change of the potentiation and depression process quickly reaches the saturation state in the preceding few pulses [26]. In this work, we designed specific waveforms to achieve improved linearity in the conductance properties, as shown in Figure 7. Figure 8a shows the synaptic plasticity of the pristine devices that applied 100 consecutive non-identical SET and RESET pulses. The 50 potentiation pulses increased from 1.26 V to 1.6 V with an increasing step voltage of 6 mV. Similarly, 50 depression pulses decreased from −1.3 V to −1.35 V with a decreasing step of 1 mV. In the potentiation process, abrupt conductance transition could be observed in the beginning few pulses. In contrast, the depression (blue dots) process shows a linear conductance change. These results indicate that the dynamics of filament growth and dissolution are not identical. In the beginning of the potentiation process, the CFs were connected to the Ti TE, which resulted in an abrupt increase in conductance change. However, the rest potentiation process showed a slight conductance increasing towards saturation, and this is due to the subsequent increase in the filament diameter [27].

During the depression process, the decrease in conductance is relatively linear compared with the potentiation process, which indicates the partial/gradual and nearly linear dissolution of CFs.

Figure 8b indicates the synaptic plasticity of the plasma-treated device. The pulse amplitude of both potentiation/depression increased/decreased from ±0.7 V to ±0.95 V with the increasing/decreasing step voltage of 5 mV. Both the potentiation and depression processes show a linear conductance change, which represents the gradual growth of filaments and, similarly, the linear dissolution of CFs. The results clearly indicate that approximately 50 analog synaptic weight states can be precisely controlled according to potentiation/depression pulse inputs. Moreover, the level of conductance change in potentiation is about 1700 µs, and the value in depression is about 1680 µs, which is almost the same in both processes, which represents precise control in pattern recognition. Moreover, we also investigated the long-term repeatable potentiation and depression cycling endurance of the device applying a total of 10 consecutive non-identical pulses, as shown in Figure 9. The plasma-treated device showed reproducible synaptic characteristics with slight conductance drifting.

Then, to analyze the weight update phenomenon, the nonlinearity (NL) of conductance change was introduced [11]. NL was defined quantitatively as Equation (5),
(5)$$\mathrm{NL}=\mathrm{Max}│G_{p}\left(n\right)-G_{D}\left(n\right)│,\;\mathrm{for}\;n=1\;\mathrm{to}\;N$$
where *G_P_*(n) and *G_D_*(n) are the conductance values after the *n*th potentiation pulse and *n*th depression pulse, respectively. The conductance values are normalized to the total range from 0 to 1 during the consecutive pulses, and NL is equal to zero for a completely linear update.

Figure 10a,b shows the normalized synaptic plasticity for the potentiation and depression processes under consecutive non-identical pulses for pristine and plasma-treated devices, respectively. The results clearly show that the NL value is 0.39 at n = 25 for the pristine devices, and the NL value is 0.06 at n = 25 for the plasma-treated devices. The NL value of the plasma-treated devices is very close to the ideal linear update. Finally, the effect of the nonlinear weight update of the plasma-treated device on the off-chip training accuracy in a neural network under the designed asymmetric waveform was studied using a binary simulation architecture with the experimentally calibrated conductance variation, as shown in Figure 10b. In the learning algorithm, we considered a neural network system containing a layer of anterior neurons, a layer of posterior neurons, and a crossbar synapse array connecting the anterior and posterior neurons. Figure 11a,c indicates that in the first training cycle, the training accuracy reached 96.1% and the inferred accuracy was 95.5%. Nevertheless, after the tenth training cycle, the training accuracy reached 96.3% and the inferred accuracy was as high as 97.1%, as shown in Figure 11b,d, respectively. The simulation results predicted that the Ar plasma processed device using the designed waveform as the input signal could significantly improve the training and inference accuracy, and only 10 training cycles were needed to achieve 96.3% training accuracy and 97.1% inference accuracy.

## 4. Conclusions

High linearity Si/SiO_2_/Ti/Pt/HfO_2_/Ti/Al RRAM devices, by employing Ar plasma treatment on the HfO_2_ thin film surface, have been demonstrated and calibrated. The plasma-treated devices exhibit a symmetric conductance distribution and pulse height in both the potentiation and depression process compared with the pristine devices. For the plasma-treated devices, the nonlinearity for potentiation and the depression process are very close to ideal linearity. Furthermore, with the designed non-identical pulse waveform, the training and inference accuracy show a significant improvement, and only 10 training cycles need to achieve 96.3% training accuracy and 97.1% inference accuracy.

## Figures and Tables

**Figure 1 nanomaterials-12-03252-f001:**
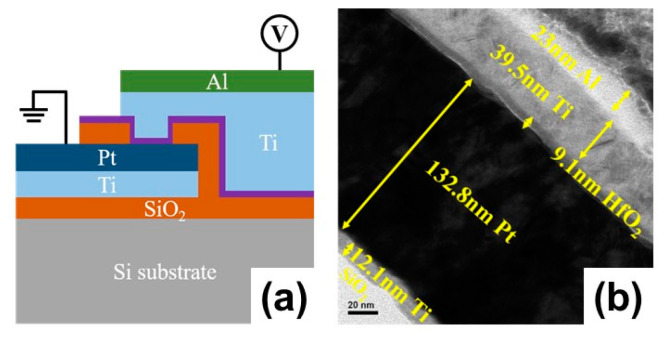
(**a**) Schematic diagram of the RRAM device with Al/Ti/HfO_2_/Pt/Ti and a (**b**) cross-section image of the TEM.

**Figure 2 nanomaterials-12-03252-f002:**
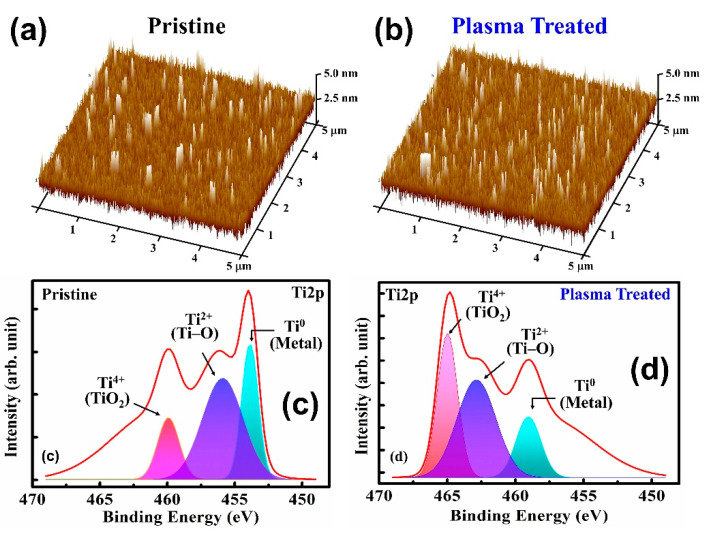
AFM topographic images of (**a**) pristine and (**b**) Ar plasma-treated HfO_2_ thin films. The Ti 2p spectra at 454.2 eV confirms the Ti^0^ valence state of the Ti metallic state peak at 459.3 eV, which corresponds to TiO_2_ bonding with the Ti^4+^ valence state of (**c**) the pristine and (**d**) plasma-treated devices.

**Figure 3 nanomaterials-12-03252-f003:**
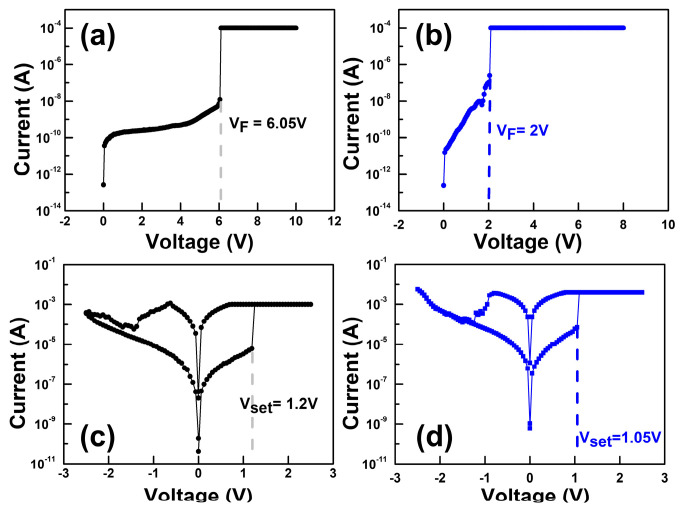
Forming curves of the (**a**) pristine and (**b**) Ar plasma-treated devices. Resistive switching characteristics of the (**c**) pristine and (**d**) Ar plasma-treated devices.

**Figure 4 nanomaterials-12-03252-f004:**
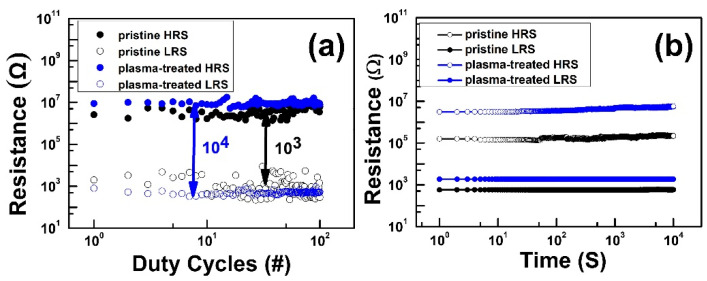
(**a**) Endurance of DC switching cycles and (**b**) retention of the pristine and plasma-treated device.

**Figure 5 nanomaterials-12-03252-f005:**
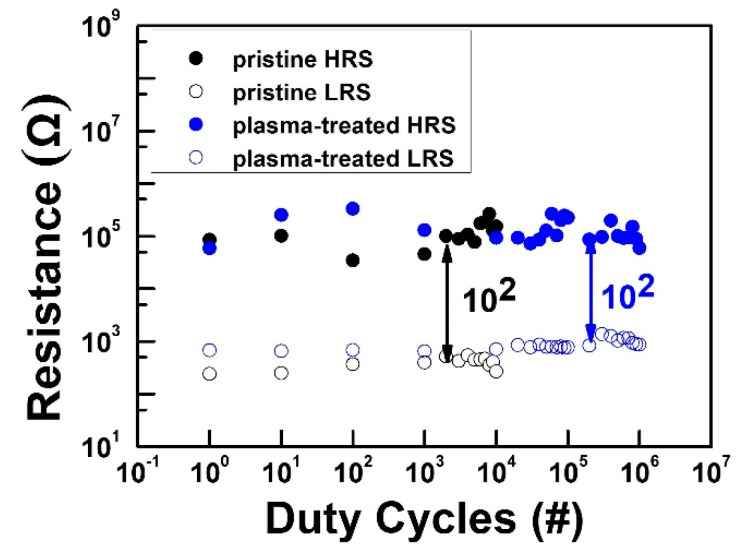
Endurance of pulse switching cycles of the pristine and plasma-treated devices.

**Figure 6 nanomaterials-12-03252-f006:**
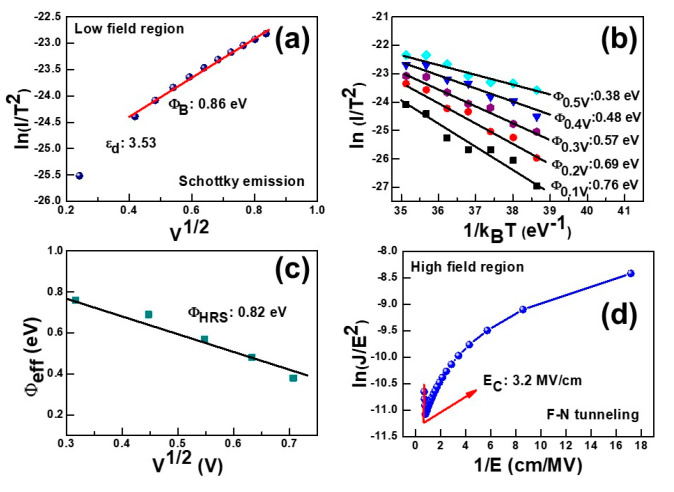
Charge transportation mechanism exploration by I–V curve fitting at room temperature. (**a**) Schottky emission [ln(*I*)–*V*^1/2^] occurred at a low field region and (**b**) temperature dependent (300 K to 330 K) Schottky emission [ln (*J*/*T*^2^)–1/*k_B_T*] in the HRS. (**c**) Schottky barrier height extraction from Φ*_eff_*–*V*^1/2^ plot in the HRS. F-N tunneling conduction occurs at *E_C_* > 3 MV/cm. *E_C_* is the critical electric field.

**Figure 7 nanomaterials-12-03252-f007:**
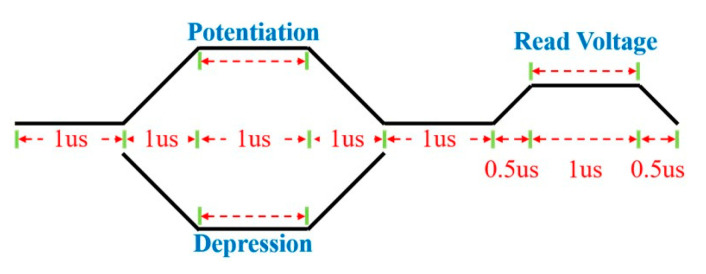
Pulse waveforms of the potentiation and depression process.

**Figure 8 nanomaterials-12-03252-f008:**
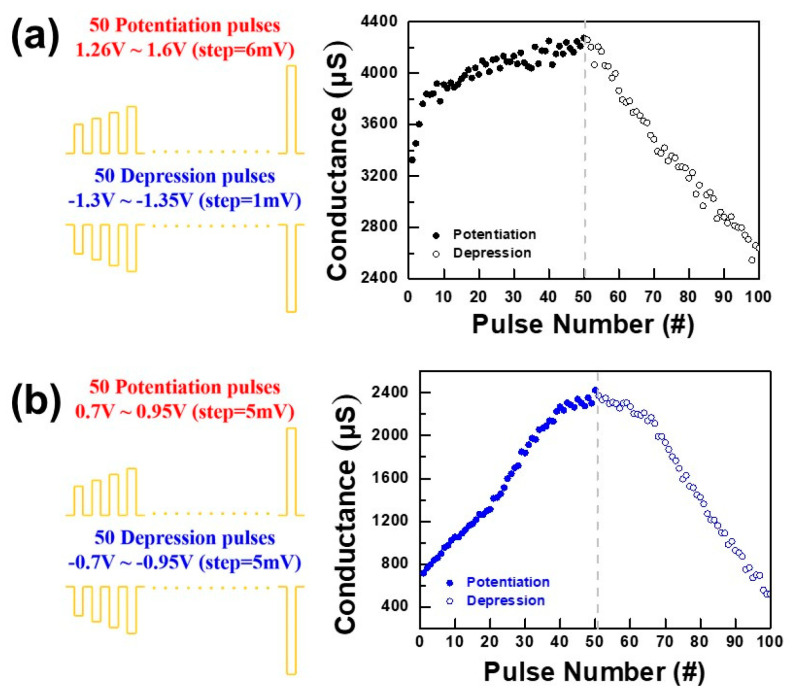
Synaptic plasticity of (**a**) pristine and (**b**) plasma-treated devices.

**Figure 9 nanomaterials-12-03252-f009:**
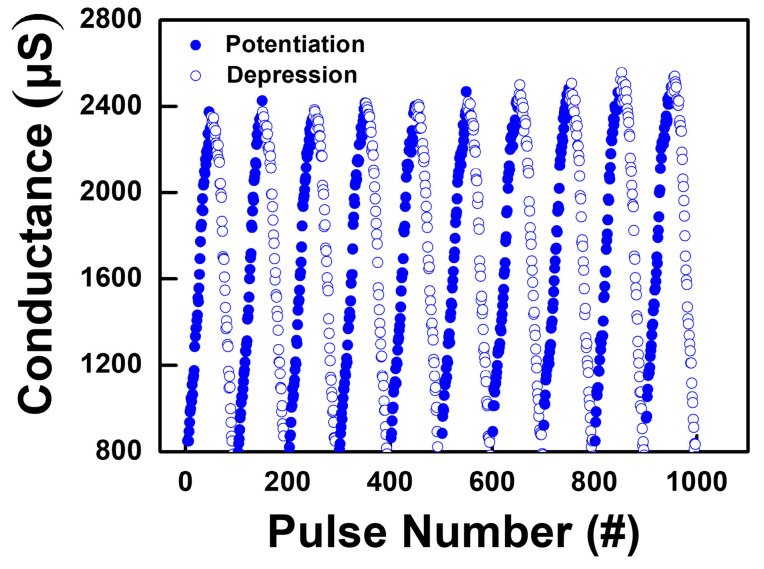
The potentiation and depression characteristics measured for 10 consecutive non-identical pulses.

**Figure 10 nanomaterials-12-03252-f010:**
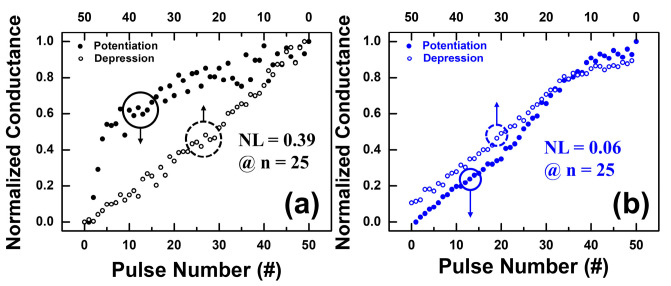
Normalized synaptic characteristics for the potentiation and depression processes under 50 consecutive non-identical pulses for (**a**) pristine and (**b**) plasma-treated devices.

**Figure 11 nanomaterials-12-03252-f011:**
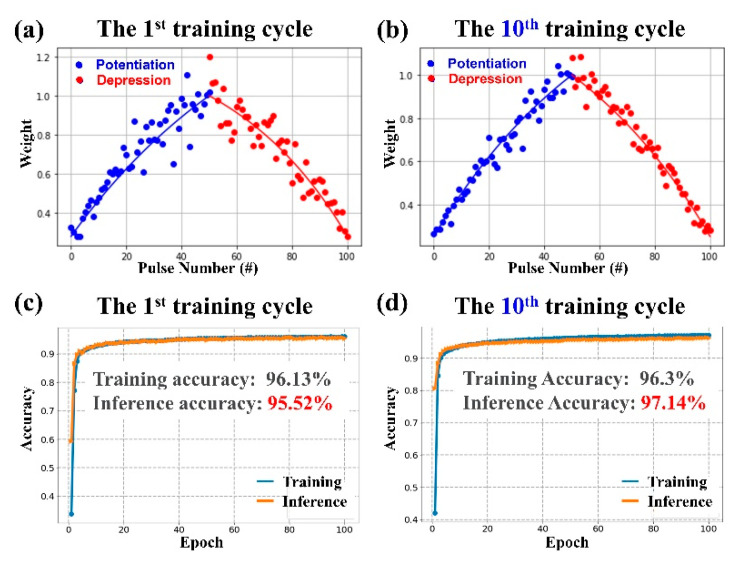
The simulated weight in (**a**) the 1st training cycle and (**b**) the 10th training cycle. The training and inference accuracy in (**c**) the 1st training cycle and (**d**) the 10th training cycle.

## Data Availability

Not applicable.

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
