# Peer review of "High Linearity Synaptic Devices Using Ar Plasma Treatment on HfO2 Thin Film with Non-Identical Pulse Waveforms"

_nanomaterials, 2022, doi:10.3390/nano12183252_

Round 1
Reviewer 1 Report
The manuscript by Lee et al. describes the improvement in synaptic devices based on Al/Ti/HfO2/Pt/Ti/SiO2/Si junction by argon plasma treatment.
In my opinion, the manuscript is interesting, and the topic fits the journal's scope. This reviewer has no particular technical remarks; however, some issues in the introduction should be improved to enhance the impact of the manuscript. Once addressed the issues listed below, I expect the manuscript to be suitable for publication in Nanomaterials.
Remarks:
The introduction does not mention the resistive switching junction that is the basis of the device. As the work is about these devices, they should mention the junction's origin and properties in the introduction. I suggest adding a paragraph.
The effect of roughness on resistive switching has been observed and investigated in several papers (see, for example, Microelectronic Engineering 126 (2014) 169–172), including different systems (see, for example, Adv. Mater. 2012, 24, 1197-1201). This evidence supports the author's thesis; thus, this literature, or similar, should be cited in the text.
Author Response
Reviewer 1
The manuscript by Lee et al. describes the improvement in synaptic devices based on Al/Ti/HfO2/Pt/Ti/SiO2/Si junction by argon plasma treatment.
In my opinion, the manuscript is interesting, and the topic fits the journal's scope. This reviewer has no particular technical remarks; however, some issues in the introduction should be improved to enhance the impact of the manuscript. Once addressed the issues listed below, I expect the manuscript to be suitable for publication in Nanomaterials.
Remarks:
The introduction does not mention the resistive switching junction that is the basis of the device. As the work is about these devices, they should mention the junction's origin and properties in the introduction. I suggest adding a paragraph.
The effect of roughness on resistive switching has been observed and investigated in several papers (see, for example, Microelectronic Engineering 126 (2014) 169–172), including different systems (see, for example, Adv. Mater. 2012, 24, 1197-1201). This evidence supports the author's thesis; thus, this literature, or similar, should be cited in the text.
Reply
Thank you for your valuable suggestions. This part has been modified in the introduction section and with color marked. Also, the literatures have been carefully cited in the revised manuscript. [14-15]
- Hee-Dong Kim, Min Ju Yun, Seok Man Hong, Ju Hyun Park, Dong Su Jeon, and Tae Geun Kim, “Impact of roughness of bottom electrodes on the resistive switching properties of platinum/nickel nitride/nickel 1 × 1 crossbar array resistive random access memory cells,” Microelectronic Engineering, vol. 126, pp. 169-172, 2014. DOI: 1016/j.mee.2014.07.018
- Massimiliano Cavallini, Zahra Hemmatian, Alberto Riminucci, Mirko Prezioso, Vittorio Morandi, and Mauro Murgia, “Regenerable Resistive Switching in Silicon Oxide Based Nanojunctions,” Mater., vol. 24, pp. 1197-1201, 2012.DOI: 10.1002/adma.201104301

Reviewer 2 Report
Authors have studied uniformity on the synaptic properties of Ar plasma treated ALD HfO2 switching layers. Some comments given below to improve the manuscript which will be beneficial for the readers in the field.
1. Introduction part can be modified to indicate more clearly the target of this experiment.
2. Some changes in XPS analysis are needed. TiOx can not be indicated as Ti4+. Only TiO2 can be denoted as Ti4+ which is the fully oxide form of Ti.
3. XPS profile for pristine and plasma treated HfO2/Ti interface looks like a mirror image. Please comment on that.
4. Can the authors present at least 100 cycles of DC I-V switching cycles together to show the clear difference between pristine and plasma-treated samples.
5. The endurance properties of both samples make confusion. Authors are requested to present both endurance properties compared in a same plot to see the difference clearly. Can be presented in the supporting information.
6. During the synaptic properties can the authors explain for considering different pulse voltage range for non-identical pulse sequence?
7. Figure 9 caption must be changed as 10 consecutive potentiation/depression cycles rather than 10^3 consecutive non-identical pulses.
Author Response
Reviewer 2
Authors have studied uniformity on the synaptic properties of Ar plasma treated ALD HfO2 switching layers. Some comments given below to improve the manuscript which will be beneficial for the readers in the field.
- Introduction part can be modified to indicate more clearly the target of this experiment.
Reply
Thank you for your valuable comments. This part has been modified in the introduction section and with color marked.
- Some changes in XPS analysis are needed. TiOxcan not be indicated as Ti4+. Only TiO2 can be denoted as Ti4+ which is the fully oxide form of Ti.
Reply
Thank you for your comment and we are sorry for the reluctant mistake. We have changed the statement as per your suggestion. The XPS peak at 454.2 eV confirms the Ti0 valency of the Ti metallic state and the peak around 455.4 eV defines the Ti2+ valence state to form the TiO and titanium suboxides, while 459.3 eV corresponds to the formation of TiO2 bonding with Ti4+ valence state at the switching layer and Ti buffer layer interface [16, 17]. It is clearly shown that the intensity has been significantly increased after the plasma treatment due to formation of thick TiO2 layer.
- XPS profile for pristine and plasma treated HfO2/Ti interface looks like a mirror image. Please comment on that.
Reply
Thank you for your comments. The Ti2p spectra in figure 2(c) of pristine device specified the large amount of Ti metallic state and a small amount of TiO2 at the HfOx/Ti interface before plasma treatment. After the plasma treatment, the roughness of the HfOx has been increased to 0.704 nm from the value 0.547 (device without plasma treatment). As a result, the greater number of Ti metallic ions oxidized at the interface and form a thick TiO2 layer at the interface due to its reactive properties as shown in the figure 2(d). So, the intensities of Ti metallic state have been decreased and the TiO2 layer has been increased in the plasma treated devices compared to the pristine devices, randomly produces as the mirror image to each other.
- Can the authors present at least 100 cycles of DC I-V switching cycles together to show the clear difference between pristine and plasma-treated samples.
Reply
Thank you for your comments. We have refitted the 100 cycles of DC I-V switching cycles data accordingly as following for Fig. 1.
Fig.1 Endurance of DC switching cycles of pristine and plasma-treated device.
- The endurance properties of both samples make confusion. Authors are requested to present both endurance properties compared in a same plot to see the difference clearly. Can be presented in the supporting information.
Reply
Thank you for your comments. We have revised the following statements (Pages 4) and Fig. 5: “Figure 5 show the P/E endurance results of the pristine and plasma-treated devices, respectively temperature at read voltage (Vread) of 0.1 V in both HRS and LRS during the pulse endurance test. Figure 5. Endurance of pulse switching cycles of pristine and plasma-treated device.”
Figure 5. Endurance of pulse switching cycles of pristine and plasma-treated device.
- During the synaptic properties can the authors explain for considering different pulse voltage range for non-identical pulse sequence?
Reply
Thank you for your comments. Considering the SET and RESET voltage limit (± 1 V), we have considered the maximum voltage amplitude less than this voltage (± 0.9 V) to avoid the thicker filament formation and also to increase device reliability. We have chosen the gradual increment/decrement (non-identical pulses) in the SET and RESET voltage to control the filament formation and rupture during the switching cycles for reliable performance and improved synaptic linearity. We also considered the identical SET and RESET pulse for the calibration of synaptic properties but couldn’t get the reliable device performance. So, we have considered the non-identical pulse scheme for reliable device performance.
- Figure 9 caption must be changed as 10 consecutive potentiation/depression cycles rather than 10^3 consecutive non-identical pulses.
Reply
Thank you for your comments. Figure 9 caption has been modified as shown in the manuscript with color marked.
